# Using a Deep Quantum Neural Network to Enhance the Fidelity of Quantum Convolutional Codes

Hanwei Xiao [ID], Xiaoguang Chen *[ID] and Jin Xu [ID]

Department of Communication Science and Engineering, Fudan University, Shanghai 200433, China; 19210720073@fudan.edu.cn (H.X.); 19210720064@fudan.edu.cn (J.X.)
* Correspondence: xiaoguangchen@fudan.edu.cn or xgchen@fudan.ac.cn

**Abstract:** The fidelity of quantum states is an important concept in quantum information. Improving quantum fidelity is very important for both quantum communication and quantum computation. In this paper, we use a quantum neural network (QNN) to enhance the fidelity of [6, 2, 2] quantum convolutional codes. Towards the circuit of quantum convolutional codes, the target quantum state $|0\rangle$ or $|1\rangle$ is turned into entangled quantum states, which can defend against quantum noise more effectively. As the quantum neural network works better for quantum states with low dimension, we divide the quantum circuits into two parts. Then we apply the quantum neural network to each part of the circuit. The results of the simulation show that the network performs well in enhancing the fidelity of the quantum states. Through the quantum neural network, the fidelity of the first part is enhanced from 95.2% to 99.99%, and the fidelity of the second part is enhanced from 93.88% to 94.57%.

**Keywords:** fidelity; quantum stabilizer code; quantum neural network

## 1. Introduction

Quantum error correction codes are used to preserve coherent states against noise. They can effectively control operational errors and decoherence. The 9-bit quantum encoding proposed by Shor in 1995 can correct single-bit phase flip error or bit flip error [1]. On this basis, Calderbank, Shor and Steane proposed a basic quantum block code coding framework called Calderbank–Shor–Steane (CSS) code in 1996, using the idea of classical linear block error correction codes, and designed quantum error correction codes using two special classical binary error correction codes [2,3]. In the same year, Gottesman proposed the stabilizer code, which is a group-theoretical structure and associated subclass of quantum codes [4]. It provides a general framework which can generate several types of quantum error correcting codes instead of specific codes.

The stabilizer formalism is very similar to the linear codes in classical error correction. Using the concept of classical error correction, it is very useful in quantum communication by preventing quantum data from encountering noise in a quantum channel. As it plays an important role in quantum error correction, it has been developed over the decades. Quantum convolutional code is a type of error correction code that combines the stabilizer formalism and the concept of traditional convolutional codes. Chau proposed a type of quantum convolutional code based on classical convolutional codes in 1998 [5]. Then, based on the concept of traditional convolutional codes [6], quantum convolutional codes were constructed using the CSS code framework and methods of the stabilizer codes. Ollivier and Tillich first described quantum convolutional codes in 2003 using stabilizer forms and providing an explicit encoding circuit [7]. Forney proposed a tail-biting quantum convolutional code, which has a higher rate and lower decoding complexity [8,9].

A good circuit should have low complexity and high reliability. Fidelity is an important factor to evaluate the reliability of the output state. If the fidelity of the states after

the decoding circuit is low, the result is not credible. Recently, a quantum neural network was proposed to enhance the fidelity of quantum states.

In 1996, G. Toth et al. constructed a quantum cellular neural network based on quantum metacellular automata [10] using physical nearest-neighbor connections [11]. In 1998, Dr. T. Menneer first explored quantum neural networks from a multi-universe quantum theoretical viewpoint [12], and later proposed a model for constructing superposed multi-universe quantum neural networks based on this [13], and compared to various conventional neural networks, he found that the performance of quantum neural networks is superior. Subsequently, various models and algorithms such as quantum-derived neural networks [14], quantum associative storage [15], quantum dot molecular models [16], and entangled neural networks [17] were proposed. Recent studies by Verdon et al. show that quantum neural networks can learn the unknown unitary operator to process the target quantum state [18]. Based on the study in [18], Kerstin et al. proposed a quantum QNN neural network framework [19], using quantum feedforward neural networks with the trained unitary operators as quantum perceptions to assist quantum computation and enhance the fidelity of quantum states.

Consider the case where all quantum gates are noisy. To resist the effect of noise from quantum gate operators, we use quantum neural networks to optimize the input coded states. The effect of resisting quantum gate noise is achieved by learning the unknown unitary operator and applying it to the coded quantum state, thus improving the fidelity of the coded quantum state before entering the quantum noise channel. We provide a simulation to show the effect of applying a quantum neural network on quantum convolutional codes.

## 2. Basic Concepts

We hope to enhance the fidelity of quantum convolutional codes with a quantum neural network in order to ensure the reliabiliy of the code. In this section, we will introduce the concepts of quantum convolutional code and a quantum neural network.

### 2.1. Quantum Convolutional Codes

Quantum convolutional codes use the stabilizer formalism which was first proposed by Gottesman [20,21]. The error syndrome is defined by the Pauli matrix.

$$\mathbf{X} = \begin{bmatrix} 0 & 1 \\ 1 & 0 \end{bmatrix}, \mathbf{Y} = \begin{bmatrix} 0 & -i \\ i & 0 \end{bmatrix}, \mathbf{Z} = \begin{bmatrix} 1 & 0 \\ 0 & -1 \end{bmatrix} \tag{1}$$

Pauli matrix $\mathbf{X}$ refers to bit flipping, $\mathbf{Z}$ refers to phase flipping and $\mathbf{Y}$ refers to both. The code subspace $\mathcal{C}$ of any stabilizer code is defined as the largest subspace stabilized by an Abelian group $S$ acting on the N physical qubits of the code. In practice, $S$ is a subgroup of the multiplicative Pauli group $G_N = \mathrm{sp}\{I, X, Y, Z\}^{\otimes N}$ [22]. The code subspace $\mathcal{C}$ can be defined by a set of independent generators $\{M_i\}$ of $S$:

$$\forall i, |\psi\rangle = M_i |\psi\rangle, |\psi\rangle \in \mathcal{C} \tag{2}$$

An $[n, k, m]$ stabilizer code is possible to encode k qubits of quantum information in n qubits and correct at most $(m - 1)/2$ errors. Similarly, the stabilizer group $\mathcal{S}$ for an $[n, k, m]$-convolutional code can be given by:

$$S = \mathrm{sp}\left\{ M_{j,i} = I^{\otimes j \times n} \otimes M_{0,i}, 1 \le i \le n - k, 0 \le j \right\} \tag{3}$$

where $M_{0,i} \in G_{n+m}$. Above, the $M_{j,i}$s are required to be independent and to commute with each other. The encoding circuit maps the to-be-protected qubits $c_{j,i}$ onto the code subspace, written as follows:

$$\left| c_{0,1}, \ldots, c_{q-1,k} \right\rangle \rightarrow \left( \prod_{i,j} \frac{1 + M_{j,i}}{\sqrt{2}} \right) \prod_{r,s} \bar{X}_{s,r}^{c_{s,r}} \left| 0, \ldots, 0 \right\rangle \tag{4}$$

for $c_{s,r} \in \{0,1\}, 0 < i \le n - k, 0 \le j < q + \lambda, 0 \le s < q$ and $1 \le r \le k$. The $\bar{X}_i$ operator is the encoded Pauli operator X. This operation can be decomposed in two steps. The first one, $\prod_{r,s} \bar{X}_{s,r}^{c_{s,r}}$ applies the different flip operators depending on the value of the to-be-protected qubits in the computational basis. The second projects this state onto the code subspace.

Take the [5,1,2] convolutional code, for example [23]. The generator matrix of the stabilizer group can be written as:

$$M = \left( \begin{array}{ccccc|ccccc} 0 & 1 & 1 & 0 & 0 & 1 & 0 & 0 & 1 & 0 \\ 0 & 0 & 1 & 1 & 0 & 0 & 1 & 0 & 0 & 1 \\ 0 & 0 & 0 & 1 & 1 & D & 0 & 1 & 0 & 0 \\ D & 0 & 0 & 0 & 1 & 0 & D & 0 & 1 & 0 \end{array} \right) \tag{5}$$

From the generator matrix, it is easy to obtain the encoding circuit as Figure 1 [23].

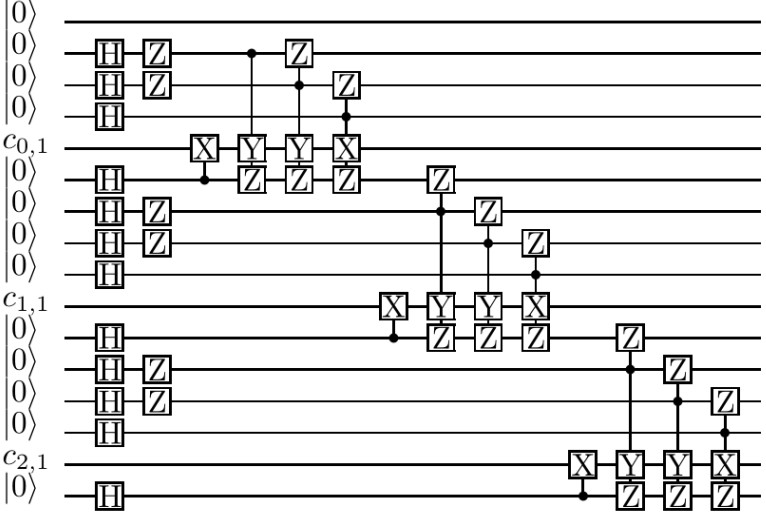

**Figure 1.** Circuit for encoding the first three qubits of a stream of quantum information with the 5-qubit convolutional code.

To estimate the error at the receiver, the quantum Viterbi algorithm was introduced. The algorithm examines the syndromes block by block and updates a list of error candidates, among which one of them coincides with the most likely error. Through the algorithm, the most likely error can be selected and then be corrected.

### 2.2. Quantum Neural Network

After decades of development, quantum neural networks have an important role in quantum pattern recognition, overcoming quantum noise, and improving the fidelity of quantum states.

The QNN quantum neural network implements a true quantum analogue of classical neurons, which form quantum feedforward neural networks capable of general quantum computation. The effective training of these networks is described in neural network models using fidelity as a cost function, providing both classical and efficient quantum implementations. This approach allows for fast optimization with reduced memory requirements: the number of quantum bits required is scaled only according to the width,

allowing for deep network optimization. The neural network has significant generalization behavior for learning unknown unitary operators and is highly robust to noisy training data with low-dimensional quantum states.

The minimal building block of QNN is a quantum perceptron, which is a quantum analogue of the perceptron used in classical machine learning. A quantum perceptron is an arbitrary unitary operator with $m$ input qubits and $n$ output qubits. These $(m + n)$ arbitrary qubits depend on $(2^{m+n})^2 - 1$ parameters. The input qubits are initialized in the possibly unknown mixed state $\rho^{\text{in}}$, and the output qubits are in the baseline product state $|0 \cdots 0\rangle_{\text{out}}$. Take the case of perceptrons acting on m input qubits and one output qubit, i.e., there are $(m + 1)$ unitary operators. A QNN is a quantum circuit composed of quantum perceptrons that are organized into $L$ hidden layers that act on the initial input state $\rho^{\text{in}}$ and usually produce a mixed output state $\rho^{\text{out}}$, with the following expression:

$$\rho^{\text{out}} \equiv \text{tr}_{\text{in, hid}} \left( \mathcal{U} \left( \rho^{\text{in}} \otimes |0 \cdots 0\rangle_{\text{hid,out}} \langle 0 \cdots 0| \right) \mathcal{U}^\dagger \right) \tag{6}$$

where $\mathcal{U} \equiv U^{\text{out}} U^L U^{L-1} \cdots U^1$ is the QNN circuit and $U^l$ is the overall unitary operator of the $l$-th hidden layer, consisting of the product of quantum perceptrons acting on the $(l - 1)$-th layer and the $l$-th layer. Since the quantum perceptrons are independent unitary operators, they do not commute with each other and need to be operated in order. The overall QNN structure is shown in Figure 2; from left to right are the input layer, hidden layer and output layer, respectively.

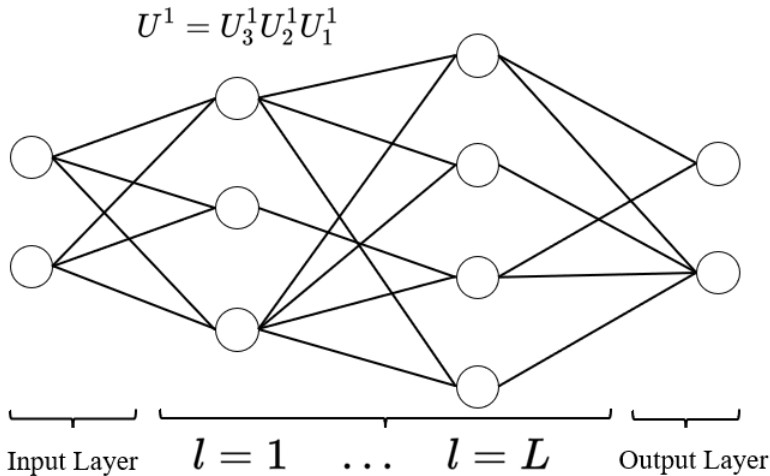

**Figure 2.** The structure of a QNN circuit.

The quantum neural network has an input layer, an output layer, and L hidden layers, where perceptron units are applied layer by layer from top to bottom. The input quantum states are expanded in the hidden layer to fit a sufficient number of perceptron units, and then we reduce the dimension to keep them consistent with the number of input qubits before reaching the output layer. A key feature of a QNN is that the network output can be represented as a combination of a series of fully positive definite layer-to-layer transition mappings $\mathcal{E}^l$, with the following expression:

$$\rho^{\text{out}} = \mathcal{E}^{\text{out}} \left( \mathcal{E}^L \left( \dots \mathcal{E}^2 \left( \mathcal{E}^1 \left( \rho^{\text{in}} \right) \right) \dots \right) \right) \tag{7}$$

$$\mathcal{E}^l \left( X^{l-1} \right) \equiv \text{tr}_{l-1} \left( \prod_{j=m_i}^{1} U_j^l \left( X^{l-1} \otimes |0 \cdots 0\rangle_l \langle 0 \cdots 0| \right) \prod_{j=1}^{m_l} U_j^{l\dagger} \right) \tag{8}$$

where $U_j^l$ is the *j*-th perceptron acting on the $(l-1)$-th and *l*-th layers, and $m_l$ is the total number of perceptrons acting on the $(l-1)$-th and *l*-th layers. The information is propagated from the input to the output, thus naturally implementing the quantum feedforward neural network.

The QNN is used to train *N* pairs of data pairs $\left(\left|\phi_x^{\text{in}}\right\rangle, \left|\phi_x^{\text{out}}\right\rangle\right)$, $x = 1, 2, \dots, N$. For a particular *x*, we train it with multiple samples to overcome the effect of quantum noise in the evaluation of the cost function. In addition, the number of samples required for each training round grows rapidly with the number of neurons, which is linearly related to the number of network parameters, making it difficult to handle quantum states with high dimension.

To evaluate the performance of a QNN quantum neural network in learning training data, a cost function needs to be constructed. In quantum information, there is an intrinsically unique measure of fidelity for the proximity of different quantum states. Defining the cost function as the fidelity between the output of the QNN and the desired output on the training data:

$$C = \frac{1}{N} \sum_{x=1}^{N} \left\langle \phi_x^{\text{out}} \left| \rho_x^{\text{out}} \right| \phi_x^{\text{out}} \right\rangle \tag{9}$$

We train the QNN by optimizing the cost function C. In each training step, the perceptron unitary operator is updated according to $U \rightarrow e^{i\epsilon K} U$, where K is a matrix containing all parameters of the corresponding perceptron unit, and $\epsilon$ is the choice step length. The matrix K is chosen such that the cost function grows the fastest. The change in cost function *C* is given by:

$$\Delta C = \frac{\epsilon}{n} \sum_{x=1}^{n} \sum_{l=1}^{L+1} \text{tr}\left( \sigma_x^l \Delta \varepsilon^l \left( \rho_x^{l-1} \right) \right) \tag{10}$$

$$\rho_x^l = \mathcal{E}^l \left( \cdots \mathcal{E}^2 \left( \mathcal{E}^1 \left( \rho_x^{\text{in}} \right) \right) \cdots \right) \tag{11}$$

$$\sigma_x^l = \mathcal{F}^{l+1} \left( \cdots \mathcal{F}^L \left( \mathcal{F}^{\text{out}} \left( \left| \phi_x^{\text{out}} \right\rangle \left\langle \phi_x^{\text{out}} \right| \right) \right) \cdots \right) \tag{12}$$

## 3. Methods

When we encode quantum states with quantum gates, the quantum gates themselves introduce noise. Quantum gates have been shown to be fault-tolerant in quantum coding. However, considering that the subsequent encoding will pass through a noisy channel, the impact of quantum gates cannot be ignored.

After we encode the quantum states with the quantum convolutional code, the fidelity of the encoded quantum states needs to be improved before entering the quantum channel due to the presence of noise in the quantum gates themselves and the fact that the quantum states involved in the encoding do not necessarily have a high purity. The QNN can then play its role.

We chose the [6, 2, 2] quantum convolutional codes. The generator matrix is as follows:

$$M = \begin{pmatrix} 1 & 0 & 0 & 1 & 0 & 0 & 0 & 1 & 1 & 0 & 0 & 0 \\ 0 & 1 & 1 & 0 & 0 & 0 & 1 & 1 & 1 & 1 & 0 & 0 \\ D & 0 & 0 & 1 & 0 & 0 & 0 & 0 & 0 & 0 & 1 & 1 \\ 0 & 0 & 0 & 0 & 1 & 1 & D & 0 & 0 & 1 & 1 & 1 \end{pmatrix} \tag{13}$$

The corresponding quantum circuit is shown in Figure 3.

As shown in Figure 3, $\left| \delta_i \right\rangle$ refers to the target quantum bit that needs to be encoded. The ratio of the number of input auxiliary qubits to the number of target qubits is 2:1. The output 6-bit quantum state is controlled by 4 qubits at the current moment and 2 qubits in the previous coding module, so that the goal of associating the input information from different moments before and after together is achieved, and the noise affected by the coding states at different moments is no longer independent and more conducive to decoding. Moreover, each 2 target qubits are in fact controlled by 6 qubits at the same time, but the

ratio of the number of auxiliary qubits to the number of encoding qubits is 2:1, which also saves some resources and achieves higher bit rates.

In order to reduce the effect caused by the noise of quantum gates, the encoded quantum state is trained with a quantum neural network (QNN). When encoding the target quantum state, the quantum state information is known, so the fidelity of the encoded quantum state information is also known before entering the quantum channel, and it can be improved by the quantum neural network. Firstly, according to the encoding circuit, we can derive the four encoded states corresponding to the four inputs, $|00\rangle$, $|01\rangle$, $|10\rangle$, $|11\rangle$, respectively, and train them with a certain volume of sample set to get the QNN for different coding states. The trained QNN is then applied to the noisy coding quantum states to obtain the high-fidelity encoded quantum states.

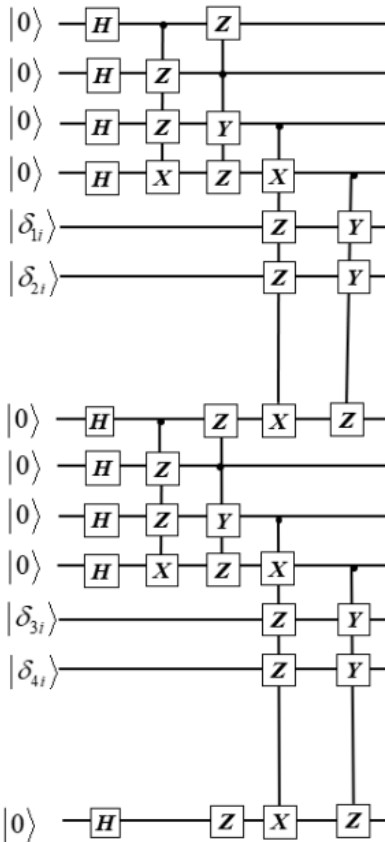

**Figure 3.** The encoding circuit of the [6, 2, 2] quantum convolutional code.

The complexity of the QNN increases exponentially with the increase in the number of qubits. To reduce the complexity and training difficulty of the QNN, we can perform block processing on the quantum convolutional code circuit, and only train 4 qubits once. The block method is as shown in Figure 4.

For each 6-bit quantum state, it is divided into two parts for the QNN training. As shown in Figure 4, the input of the 4-bit quantum state in the first part is $|0000\rangle$, and the first 4 qubits of the encoded state are obtained after passing through the encoding circuit, which is not affected by the input information, so 4 different input quantum states can all use the same QNN. The 4-bit quantum state in the second part is encoded by $|00\rangle$ through the quantum gate operation, and then the input quantum state is controlled for encoding. Therefore, 4 different QNNs are required to be trained for 4 input quantum states.

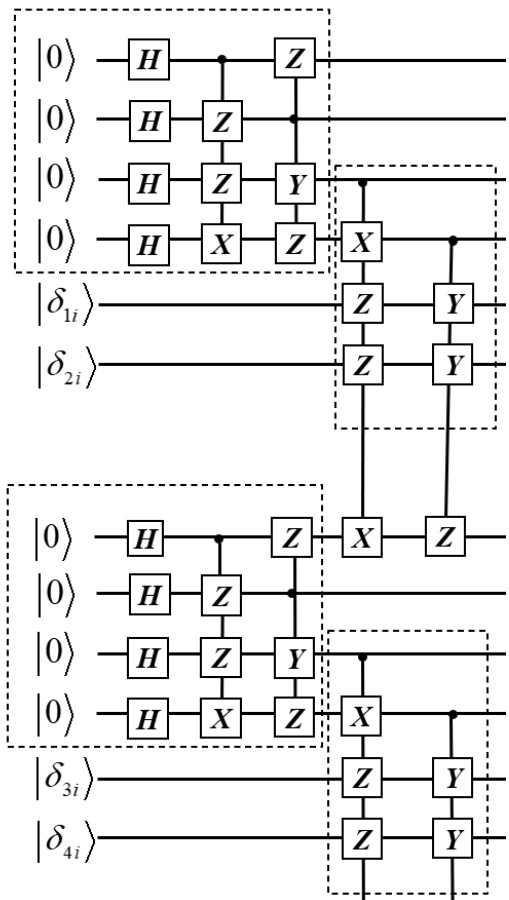

**Figure 4.** The block method of the [6, 2, 2] quantum convolutional code.

Take the input quantum state $|00\rangle$ as an example. We construct two QNNs with the expectation that the quantum states are trained to the standard quantum states after noiseless quantum gates. For the first part of the quantum state, the expected result is $\frac{1}{4}(|0000\rangle + |0001\rangle + |1000\rangle + |1001\rangle)$, and the expected result of the second quantum state is $\frac{1}{4}(|0000\rangle + |0011\rangle + |0100\rangle + |0111\rangle - |0001\rangle - |0010\rangle - |0101\rangle - |0110\rangle)$. In this way, the target output quantum state is known, so the unknown unitary operator can be trained by using the sample sets, and the resulting QNN can be trained layer by layer according to the input encoded quantum state to obtain the unitary operator that adapts the noisy encoded quantum state. Then, we use the unitary operators acting on the encoded quantum state to obtain a quantum state that approximates the target noiseless quantum state. The settings of the two QNNs are identical except that the sample set and the target quantum state are different.

## 4. Simulation and Analysis

Assuming that all quantum gates are noisy quantum gates, we provide the simulation of the whole circuit in Figure 5.

It can be seen through simulation that when the noise of the quantum gate is taken into account, the qubit error rate is greatly increased. As the SNR increases, in the case of a noisy quantum gate, the qubit error rate interval is [23.44%, 0.45%], and compared with the noiseless quantum gate, the qubit error rate interval is [14.96%, 0.057%].

The results of training the first part of the quantum state are shown in Figure 6. Without the QNN, its fidelity is relatively discrete. After applying the QNN on the quantum state, the fidelity of the quantum state is very close to 1, indicating that it is very close to the desired target quantum state.

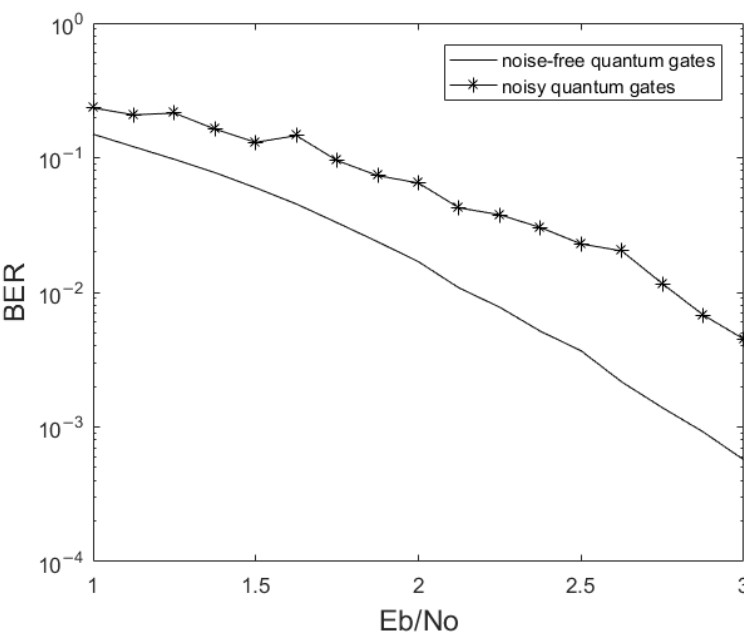

**Figure 5.** Simulation of the [6,2,2] quantum convolutional code with noisy quantum gates.

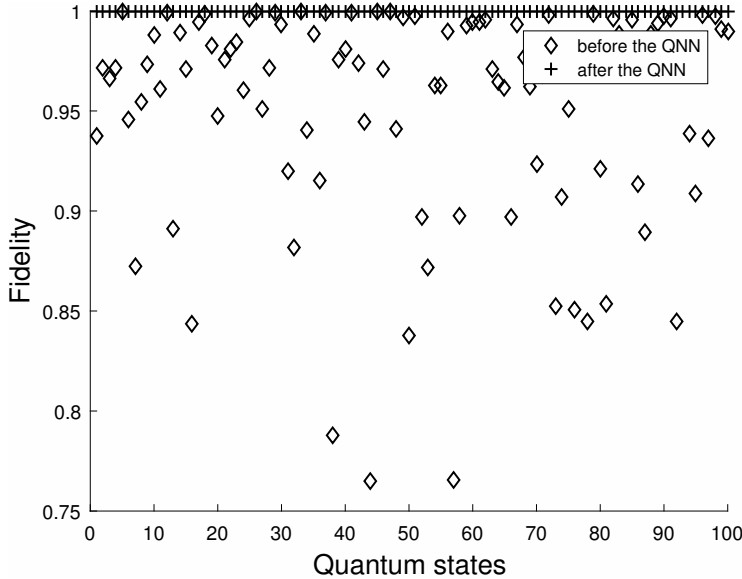

**Figure 6.** The fidelity of quantum states before and after training the QNN on the quantum state of the first part.

However, the training effect of the second part is relatively poor. It can be seen from Figure 7 that the fidelity is closer to 1 than before training, but it is still relatively discrete. By observing the quantum states of the two parts, it is not difficult to see that the quantum states of the second part have a higher degree of entanglement and are more complicated to train, so the fitting may be poor.

We simulate the cost function of the two parts to observe the fit of the QNN in Figure 8.

As shown in the figure, with the increase of training samples, the cost functions of both parts rise rapidly, but the trends are different. The cost function of the first part converges quickly, the rise is linear, and the final convergence value is very close to 1, so the training effect is better, and the fidelity close to 1 can be obtained when applied to the encoding quantum state. The second part of the cost function is more tortuous in ascending, probably because the data is more complex, it is difficult to select a suitable

direction, and the convergence is slower. After fitting, it finally converges to 0.92, which is still some distance from 1. The fitting effect is not ideal, but still useful for improving the fidelity of the quantum state.

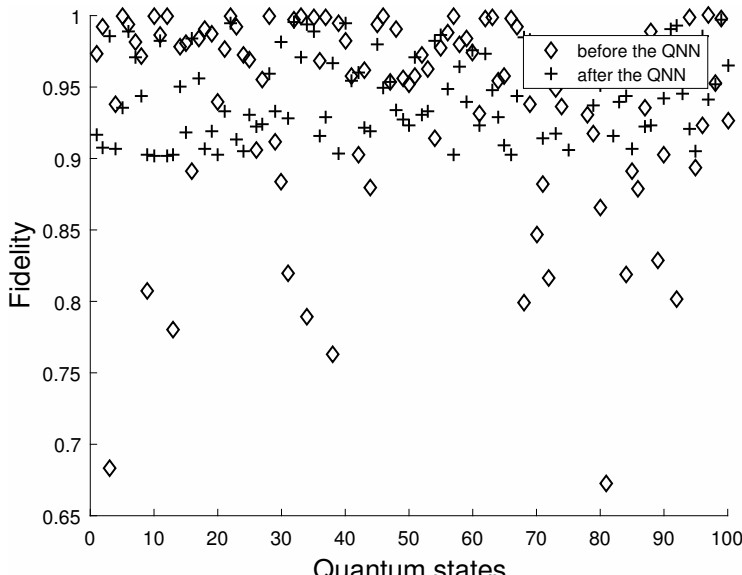

**Figure 7.** The fidelity of quantum states before and after training the QNN on the quantum state of the second part.

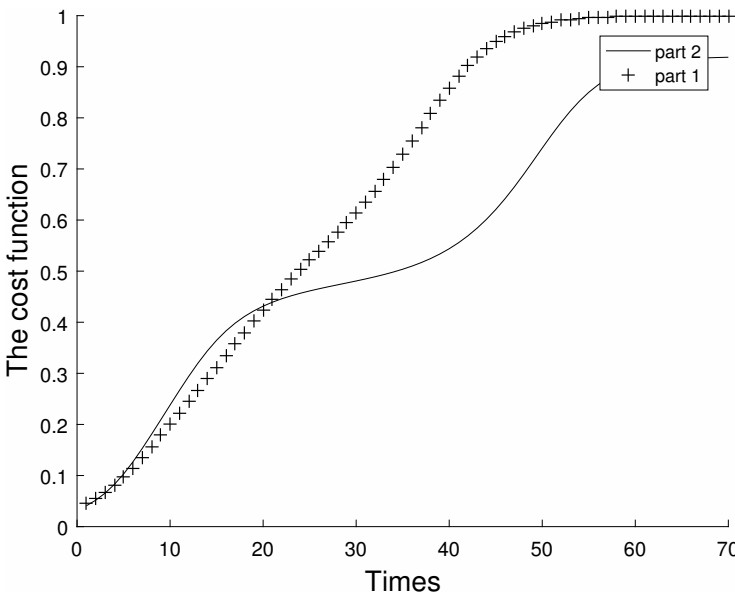

**Figure 8.** The cost function for the QNNs of two parts.

After applying the QNN for all encoded quantum states, the trained encoded quantum states are used to enter the quantum noise channel. We simulate the final BER, and the result is shown in Figure 9.

As shown in the figure, the final BER after QNN is intermediate between the case of noiseless quantum gates and the case of noisy quantum gates. Since the convergence of the second part of the QNN is not sufficient, the encoded quantum state cannot reach the noise-free criterion. In addition, the quantum gate at the receiver used to decode the encoded quantum state is also noisy. Since the received quantum information cannot be obtained at the receiver side without measurement, the QNN cannot be applied to optimize the decoding circuit at the receiver.

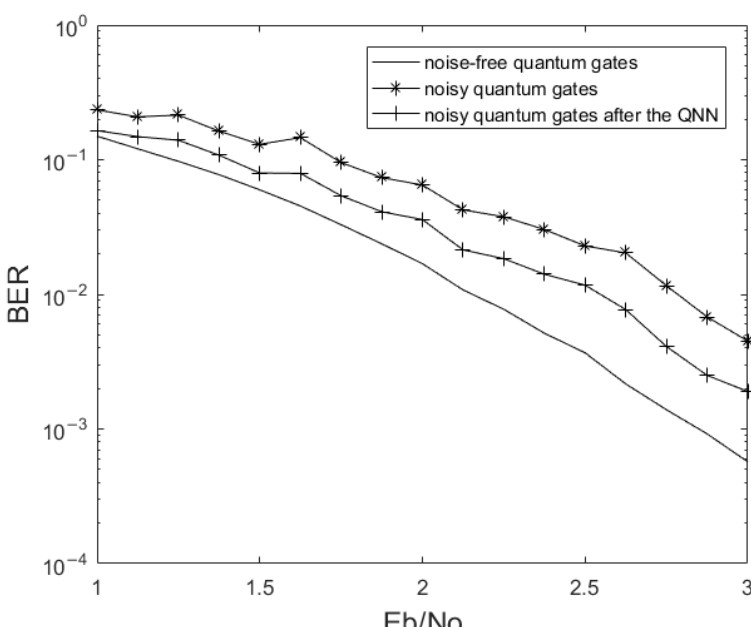

**Figure 9.** The BER of the [6, 2, 2] quantum convolutional codes applying the QNN compared with BER before training and the noiseless quantum gate.

In a word, the quantum neural network has a good correction effect on the encoding state of the quantum convolutional code. It can defend the noise of quantum gates by improving the fidelity of the encoded quantum state, which has a certain effect on reducing the BER.

## 5. Conclusions

This paper provides a method to reduce the error probability of the quantum convolutional code by enhancing the fidelity of the encoded states using quantum neural networks. Considering that the quantum gates used by the encoding and decoding circuits have noise, we use a quantum neural network to optimize the quantum state after the encoding circuit. A quantum neural network can convert a quantum state with lower fidelity into a target quantum state with higher fidelity through the unitary transformation obtained by learning. Through quantum neural networks, the fidelity of a large number of noisy quantum states is improved from a discrete state to a state whose fidelity is close to 1.

The increase in qubits to be processed will bring exponentially increased complexity to the quantum neural network. When too many qubits are input to the QNN once, the gradient will collapse, and the fidelity cannot be effectively improved. Therefore, we divide the encoding circuits that need to be optimized for processing, and divide the quantum state with more qubits into multiple quantum states with fewer qubits for quantum neural network optimization. After the quantum neural network optimization, the BER is greatly improved compared with the unoptimized case. After simulation, under the condition of 3 dB SNR, the BER of 0.45% in the case of noisy quantum gates can be reduced to 0.19%.

In the case of assuming no noise, the signal-to-noise ratio at 3 dB is 0.057%. There is still a certain gap after quantum neural network optimization. The reasons include: (1) the output quantum state of the decoding circuit is unknown and cannot pass through the quantum neural network. However, the quantum gate of the decoding circuit also has noise; (2) the training results of the quantum neural network for more complex quantum states are poor, and the effect of completely improving the fidelity to 1 cannot be achieved. Thus, there is a certain gap with the ideal situation.

**Author Contributions:** Formal analysis, H.X.; Supervision, X.C.; Writing—review & editing, J.X. All authors have read and agreed to the published version of the manuscript.

**Funding:** This research received no external funding.

**Institutional Review Board Statement:** Not applicable.

**Informed Consent Statement:** Not applicable.

**Data Availability Statement:** Not applicable.

**Conflicts of Interest:** The authors declare no conflict of interest.

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
