# Peer review of "Using a Deep Quantum Neural Network to Enhance the Fidelity of Quantum Convolutional Codes"

_applsci, doi:10.3390/app12115662_

Round 1
Reviewer 1 Report
Overall the paper is well-written and clear. The idea of combining the QNN in [18][23] with quantum error correction is interesting and novel.
The presented simulation results look encouraging, although the application to more complex quantum states appear to be challenging.
One thing I wonder is if the approach is specific to quantum convolutional codes, or if it can be applied to other codes as well. Regarding the impact of noisy gates on quantum convolutional codes (or other codes), I wonder if this has been studied before. In other words, do the authors solve some problem that has been known before?
Specific comments:
- in the last paragraph of Section 1, the connection to [18] and [23] should be mentioned
- it seems that (5) corresponds to a [5,1,2] convolutional code rather than [5,1,3]
- variables in text should always be in mathematical style
Author Response
- comment: One thing I wonder is if the approach is specific to quantum convolutional codes, or if it can be applied to other codes as well. Regarding the impact of noisy gates on quantum convolutional codes (or other codes), I wonder if this has been studied before. In other words, do the authors solve some problem that has been known before? Respond 1. We take quantum convolutional code as an example. This approach can be applied to other codes as well. The impact of noisy gates is a well known problem and several approaches have been proposed such as purification and so on. Using QNN is a new choice.
- comment: in the last paragraph of Section 1, the connection to [18] and [23] should be mentioned. Respond 2. Thanks for this suggestion. We have mentioned that in section 2, perhaps not clearly enough. We change the statement to make it clearly and move it to Section 1.“Recent studies by Verdon et al. show that quantum neural networks can learn the unknown unitary operator to process the target quantum state [23]. Based on the study in [23], Kerstin et al. propose a quantum QNN neural network framework [18], using quantum feedforward neural networks with the trained unitary operators as quantum perceptions to assist quantum computation and enhance the fidelity of quantum states.”
- comment: it seems that (5) corresponds to a [5,1,2] convolutional code rather than [5,1,3]. variables in text should always be in mathematical style. Respond 3. Thanks for this suggestion. We’re sorry for the mistake. We have corrected it in the revised version.

Reviewer 2 Report
I'd a great pleasure of reading this paper, which clearly presents the use of quantum neural network for the increasing of fidelity of error correcting codes. The qualitatively expected results are supported by numerical simulation.
In my opinion the paper can be published in its present form with very minor corrections. Namely, all bibliographic references should be completed to include not only the titles and publication dates, but also the Volumes and Pages for the journals, or the Publishers for the books.
page. 11:
line 264 , Item 23: I guess it should be ''arXiv.org: 1806.09729'', but the authors should specify what do they really mean.
line 262, Item 21: I guess the publisher should be "Elsevier"
line 252, Item 14: I guess it should be "Proceedings of NIPS'95"
line 249, item 12: Publisher and address should be indicated
line 242, item 8: same
line 239, item 6: there are some typos, and the Publisher should be indicated
line 238, item 5: volume and pages should be indicated
line 235, item 3: Journal title should be properly capitalized
Author Response
Thanks for pointing out the issue. We have corrected the mistakes in the revised version.